



# Calibration of Optical Particle Spectrometers Using Mounted Fibres

Jessica Girdwood[1,a,b], Harry Ballington[1], Chris Stopford[1], Rob Lewis[1], and Evelyn Hesse[1]

[1]Centre for Atmospheric and Climate Physics, University of Hertfordshire, Hatfield, Hertfordshire, AL10 9AB
[a]Now at: School of Earth, Atmospheric and Environmental Sciences, University of Manchester, Manchester, M13 9PL, UK
[b]Now at: National Centre for Atmospheric Science, School of Earth, Atmospheric and Environmental Sciences, University of Manchester, Manchester, M13 9PL, UK

**Correspondence:** Jessica Girdwood (jessica.girdwood@ncas.ac.uk)

**Abstract.** Calibrations of OPSs are non-trivial, and conventionally involve aerosolisation techniques which are challenging for larger particles. In this paper, we present a new technique for OPS calibration, which involves mounting a static fibre within the instrument sample area, measuring the scattering cross section, then comparing it with a calculated value. In addition, we present a case for the use of GLMT simulations to account for deviations in both minor and major axis beam intensity, which

has a significant effect on particles which are large compared to the beam waist, in addition to reducing the need for a 'top-hat' spacial intensity profile. The described technique is OPS independent and could be applied to a field calibration tool, which could be used to verify the calibration of instruments before they are deployed. In addition to this, the proposed calibration technique would be suited for applications involving mass production of low-cost OPSs.

## 1 Introduction

In-situ optical particle spectrometers (OPSs), which are also referred to as optical particle counters (OPCs), are a common class of atmospheric instrumentation, primarily utilised for diameter measurements of aerosol particles and cloud droplets. This is a commonly required parameter for the characterisation of aerosol-cloud interactions (Redemann et al., 2021), cloud evolution and precipitation dynamics (Pinsky and Khain, 2003), and aerosol-radiation interactions (Haywood et al., 2021) – all of which modulate aerosol radiative forcing, which causes large uncertainties in climate model predictions (Masson-Delmotte

et al., 2021).

OPSs utilise the optical elastic scattering properties of an aerosol particle to derive its physical parameters. Normally this [parameter] is diameter, however the Small Ice Detector (SID, Cotton et al., 2010) and Particle Phase Discriminator (PPD, Vochezer et al., 2016) Instruments utilise spacially-resolved elastic scattering to determine particle phase and morphology – which is most commonly applied to ice-crystal habit.

A generalised OPS will directly measure scattering cross section (SCS, Pinnick and Auvermann, 1979). In order to compute particle radius from SCS, an inverse problem must be solved whereby a simulation of SCS versus some physical dimension – unique to a specific material – is used to generate a lookup table or instrument response curve. This response curve has an inherent dependency on material optical properties, namely real and imaginary refractive index, physical homogeneity





and morphology, and isotropicity. Commonly, Mie-Lorenz theory is used to generate this response curve, which adds the
assumption that the aerosol consists of spherical particles.

The OPS will normally achieve a SCS measurement by illuminating the particle with a light source – commonly a laser diode – and integrating the scattered light over the area of some photodetector (normally a photo-multiplier tube or photo-diode). The exact angles chosen is a design compromise between increasing the scattered light intensity, reducing dependency on refractive index, and creating a more monotonic lookup table. For example, the Forward Scattering Spectrometer Probe
(Dye and Baumgardner, 1984; Baumgardner et al., 1985; Baumgardner and Spowart, 1990, FSSP,) uses forward scattering, which leads to a large scattered light intensity on the detector (and therefore high signal to noise ratio), and a low dependency on refractive index. However, the Universal Cloud and Aerosol Sounding System (UCASS, Smith et al., 2019) uses a detector centred on $60°$ which results in a highly monotonic lookup table for Mie-Lorenz computations.

A modern OPS is calibrated by creating a linear calibration function to map photoelectric current produced by the photode-
tector to SCS, as demonstrated in Rosenberg et al. (2012). This calibration function has two coefficients. The $0^{th}$ coefficient (y-intercept) is primarily determined by the amount of stray light on the photodetector resulting from light from the laser scattering off internal instrument surfaces, since no surface is perfectly absorbing, nor is any laser perfectly aligned. The $1^{st}$ coefficient (gradient) depends on the transimpedance amplifier (TIA) current gain, the quantum efficiency of the photodetector, and the tolerances in analogue signal processing electronics.

This calibration function is determined experimentally. Conventionally, this experiment consists of generating size-monodisperse aerosol, and transporting it through the sample area (Pinnick and Auvermann, 1979; Kim and Boatman, 1990; Rosenberg et al., 2012); well-characterised multi-modal particle sources have also been used (Binnig et al., 2007; Heim et al., 2008). However, these techniques are reliant on the generation and transportation of aerosol, which renders the calibration sensitive to both particle loss mechanisms and perturbations in particle size due to the generation apparatus. Particle losses become particularly
significant while generating aerosol particles with high inertia, due to gravitational settling and deposition in transport tubing. This is a problem because larger sizes are important for constraining the $1^{st}$ calibration coefficient. In addition, calibrations which use manufactured monodisperse silica, glass, or latex spheres – for example Rosenberg et al. (2012) – are dependent on engineering tolerances of said monodisperse particles, which leads to a tolerance stacking effect whereby the error in particle size estimate is the product of instrument tolerances and particle calibration standard tolerances.

The conventional calibrations use homogeneous spheres to generate scattered light, since the scattering can be solved analyt-ically. However, there is no reason beyond this why spheres need to be used to create a specific SCS in the beam; as long as the SCS is known exactly, any object can be used. One object from which the SCS can be determined analytically is an infinitely long fibre. If such a fibre were to be long enough to completely traverse the laser beam, it could be considered as such. In this paper, we experiment with the use of statically mounted fibres for the generation of a SCS in the calibration of OPSs.
The UCASS was chosen as the tested OPS because they are research instruments which are produced in moderately-sized batches, meaning many versions of the same instrument can be compared against eachother. It is the intention of the authors that the presented technique is OPS independent. A number of UCASS OPSs were calibrated using both the fibre method and a conventional method involving aerosol microspheres. It is the intention that this technique can be used to supplement





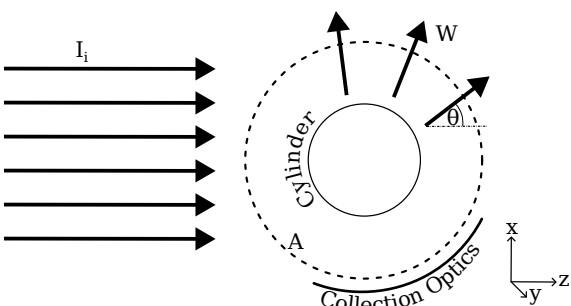

**Figure 1.** An illustration of the variable definitions used for the scattering calculations. $I_i$ is the incident irradiance, $W$ is the scattered light power which crosses far field area $A$, and $\theta$ is the zenith angle.

conventional aerosol calibrations when larger particles are considered. In addition, a static fibre tool could be used as a portable
field calibration method since it would not require a clean environment, nor a bulky aerosolisation apparatus.

## 2   Theory

### 2.1   Fibre SCS Calculation

Generally, scattering cross section can be defined from its relation to the scattered electromagnetic energy – with units of Watts – and input irradiance – with units of Watts per square metre – which is

$$\sigma = \frac{W}{I_i} \tag{1}$$

where $\sigma$ is the SCS with units of square metres, $W$ is the electromagnetic energy which crosses the surface of an imaginary cylinder in the far field (labeled $A$ in Fig. 1), and $I_i$ is the incident irradiance. The variable definitions are consistent with Fig. 1.

The SCS of an arbitrary particle, taken from Bohren and Huffman (1998), is given by

$$\sigma = \int\limits_{0}^{2\pi} \int\limits_{0}^{\pi} \frac{|\mathbf{X}|^2}{k^2} \sin\theta \, d\theta \, d\phi \tag{2}$$

where $\sigma$ is the SCS, $\theta$ is the zenith angle, $\phi$ is the azimuth angle, $k$ is the wavenumber, $\mathbf{X}$ is the vector scattering amplitude. It is related to phase function by

$$p_{11} = \frac{|\mathbf{X}|^2}{k^2 \sigma} \tag{3}$$

where $p_{11}$ is the phase function. Phase function is the distributed scattered electromagnetic radiation intensity, which is nor-
malised such that its integral is unity. The SCS of a long fibre which intersects an OPS laser beam can be treated as an infinitely



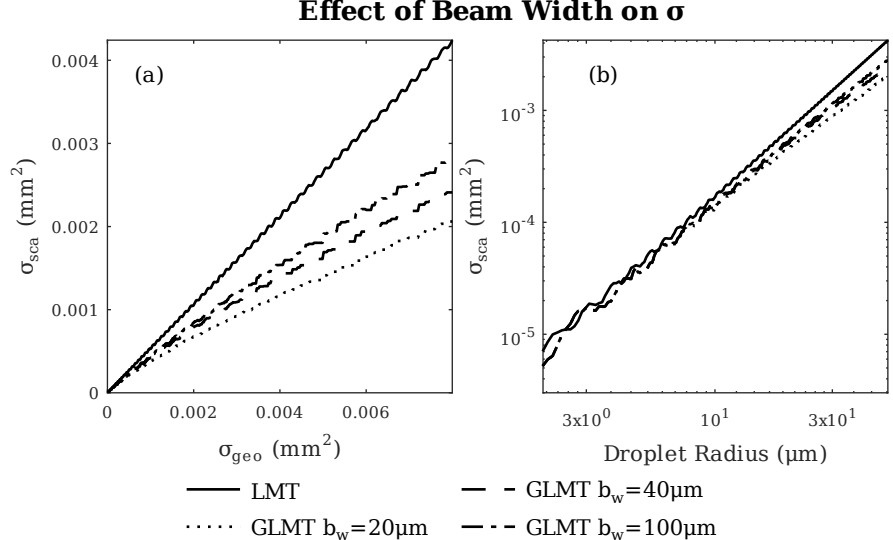

**Figure 2.** A figure containing plots of the GLMT simulation results compared to conventional LMT. The left panel, (a), shows scattering cross section versus geometric cross section for LMT and three different beam widths; the right panel, (b), shows the same, but on a log-log axis and with the geometric cross section abstracted by droplet radius – a more conventional plot for OPS users. The refractive index used was that of water (1.33+0j).

long cylinder with normally incident irradiance. Since the SCS of an infinitely long cylinder would be infinite, the SCS per unit length can be calculated (Bohren and Huffman, 1998). Thus Eq. 1 becomes

$$\sigma = \int_0^l \bar{\sigma}\, dy = \frac{W}{I_i} \qquad (4)$$

where $\bar{\sigma}$ is the SCS per unit length with units of square metres per metre, $l$ is the length of the fibre with units of metres, $y$ is the
distance in the direction of the axis of the cylinder – as shown in Fig. 1 – and the rest of the variable definitions are consistent with Fig. 1 and Eq. 1.

In the case of an OPS, the length of the fibre is actually the width of the laser beam in the $y$ direction. This means that the assumption of negligible diffraction over the flat edges of the fibre is valid, because these edges are not within the laser beam in this scenario. However, since laser beams in OPSs are often focused at the sample area, the intensity distribution of the laser
beam in the $y$ direction is a gaussian distribution. It is therefore more convenient to express Eq. 4 as

$$\sigma = \int_0^\infty \bar{\sigma}\beta(y)\, dy \qquad (5)$$



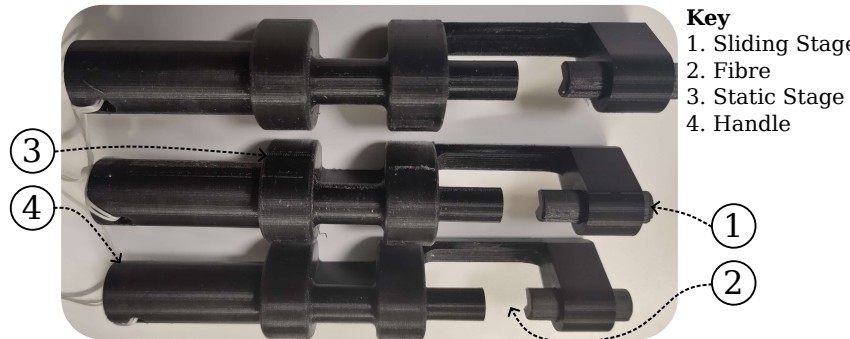

**Key**
1. Sliding Stage
2. Fibre
3. Static Stage
4. Handle

**Figure 3.** A picture of the custom designed fibre mounts which were used to insert the fibres into the UCASS laser beam for the duration of the calibration. The two stages of the fibre holder and the place for the fibre are labeled.

where $\beta(y)$ is the energy distribution of the laser beam as a function of distance along the $y$ axis, and is normalised such that its integral is equal to the width of a beam with a 'top-hat' energy distribution, and the same average energy density and total energy as the real beam. The double integral in Eq. 2 is then more convenient to express in cylindrical coordinates, thus SCS of a fibre which extends over the full width of the OPS laser is then given by

$$\sigma = \int\limits_0^\infty \int\limits_0^{2\pi} \frac{|\mathbf{X}|^2}{k^2}\,\beta(y)\,sin\,\theta\,d\theta\,dy \tag{6}$$

Since, in this instance, $\bar{\sigma}$ is not a function of $y$ because the fibre is uniform along its axis, the outer integral in Eq. 6 can be solved. For practical applications, the SCS integral needs to be modified to include a weighting function which accounts for the instrument optics, similar to that used in Pinnick and Auvermann (1979); Rosenberg et al. (2012). This has a value of either 0 or 1, and is a function of both $\theta$ and $\phi$. However – in order to dismiss the outer integral in Eq. 6 – it is more convenient to express the weighting function as representing the proportion of $\phi$ angles collected as a function of $\theta$ only. Equation 6 therefore becomes

$$\sigma = d_s \int\limits_0^{2\pi} \frac{|\mathbf{X}|^2}{k^2}\,\omega(\theta)\,sin\,\theta\,d\theta \tag{7}$$

where $\omega(\theta)$ is the weighting function representing the instrument optics, and $d_s$ is the width of the theoretical top-hat beam. The actual value of $d_s$ can be incorporated into the final calibration coefficient. If the collecting optics have a sufficiently large radius of curvature; and the laser beam thickness is sufficiently small; and the laser beam focusing optics have a sufficiently large focal length, then the change in the amount of light collected with respect to distance in the $y$ direction is negligible. Therefore, the value of $\omega(\theta)$ for an OPS with $n$ number of collecting optical surfaces can be expressed generally as

$$\omega(\theta) = \sum\limits_{i=1}^{n} \left[ H\left(\theta - \Theta_{i,1}\right) - H\left(\theta - \Theta_{i,2}\right) \right] \tag{8}$$





where $\Theta_1$ and $\Theta_2$ are the beginning and terminating angles of collecting optic number $i$, and $H$ is the Heaviside function of $\theta$. For the UCASS, the assumptions made for Eq. 8 are valid, however this would need to be carefully considered if one were to apply this equation to cloud probes with large optical path lengths. In that instance, the weighting function of $\theta$ should be re-derived since it would have values other than 0 and 1. $\dot{\sigma}$ can be computed by common Mie scattering codes, for example Schäfer (2011), which was used for the cylinder SCS computation in this paper.

## 2.2 Sphere SCS Calculation

In this work, the SCS calculation for a sphere needs to be conducted to two reasons: to compute the SCS from the traditional aerosol calibration, and to calculate the final data product, that is, cloud droplet or aerosol particle diameter from the SCS measured by the instrument. In previous works, the variation in beam power along its minor axis – the $y$ direction – has been neglected. However, for a sufficiently large particle, this assumption is invalid.

Since, in this instance, the differential SCS in Eq. 5 is a function of $y$, the outer integral in Eq. 6 cannot be solved as easily. In addition, the SCS integral is now more convenient to express in spherical coordinates, therefore the detecting optics weighting function is applied to Eq. 2. In order to solve the differential SCS as a function of $\theta$ and $\phi$, while accounting for the non uniform intensity profile of the laser beam in the $y$ direction, generalised Lorenz-Mie theory (GLMT) can be employed. GLMT works by altering the Mie coefficients of the electromagnetic waves in order to render them representative of a gaussian beam. Since this method implicitly accounts for $\beta(y)$, the SCS equation is then

$$\sigma = \int\limits_0^{2\pi} \int\limits_0^{\pi} \frac{|\mathbf{X}|^2}{k^2} \omega(\theta,\phi) \sin\theta \, d\theta \, d\phi \tag{9}$$

Note that, in this instance, the weighting function is both a function of $\theta$ and $\phi$ due to the azimuthally non-constant scattering amplitude. $\dot{\sigma}$ can be, in this instance, obtained from the GLMT simulations. A generalised, OPS independent weighting function is derived in vector form here as

$$\omega(\theta,\phi) = \begin{cases} 1 & \text{for } \angle\left(\overrightarrow{R}(\theta,\phi)\overrightarrow{C}\right) \leq \alpha_l, \\ 0 & \text{for } \angle\left(\overrightarrow{R}(\theta,\phi)\overrightarrow{C}\right) > \alpha_l \end{cases}, \tag{10}$$

$$\overrightarrow{R}(\theta,\phi) = \left\langle \cos\theta\cos\phi\,\hat{\mathbf{i}}, \cos\theta\sin\phi\,\hat{\mathbf{j}}, \sin\theta\,\hat{\mathbf{k}} \right\rangle, \tag{11}$$

$$\overrightarrow{C} = \overrightarrow{R}(\Theta_l, \Phi_l) \tag{12}$$

where $\overrightarrow{R}$ is a vector in the direction of a scattered light ray with zenith and azimuth angles of $\theta$ and $\phi$; $\overrightarrow{C}$ is a vector in the direction of the geometric centre of the surface of the collecting optic; $\hat{\mathbf{i}}$, $\hat{\mathbf{j}}$, and $\hat{\mathbf{k}}$ are unit vectors in the $x$, $y$, and $z$ directions, as labeled in Fig. 1; $\alpha_l$ is the half angle of the collecting optic; and $\Theta_l$, $\Phi_l$ are the zenith and azimuth angles of the centre of the collecting optic. If there are multiple collecting optics, or – as there often are for beam dumps – holes in any collecting optics then $\omega$ can be computed for each component – where holes in optical surfaces are treated as components – then added or subtracted as necessary.





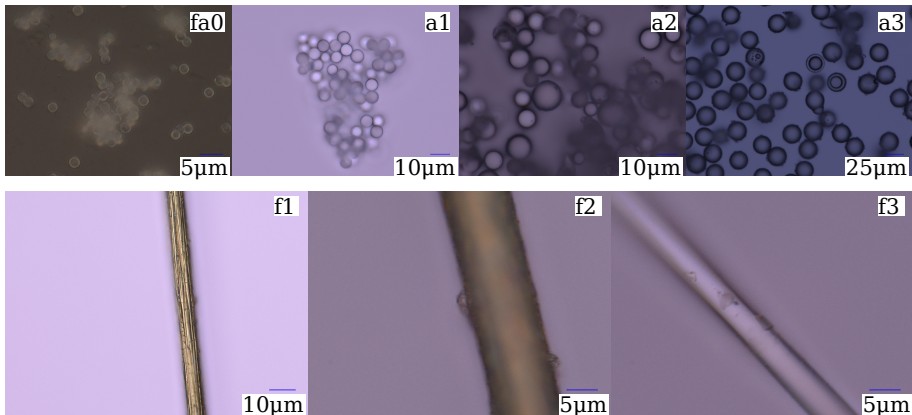

**Figure 4.** A series of optical microscope images of the aerosol particles and fibres used for the calibrations in this paper. The labeling of each image is consistent with Tables 1 and 2.

The GLMT simulations which were used in this paper are described in Jia et al. (2017), and the beam shape coefficient calculation which used was the localised approximation method described in Wang and Shen (2018). The difference that GLMT versus conventional Lorenz-Mie theory (LMT) scattering simulations made to the finished SCS lookup table is shown in Fig. 2. The discrepancy between the two calculation methods is more noticeable when the beam width is small when compared to the size of the particle. This would lead to under-sizing of larger particles, since the same detector photocurrent would be interpreted as scattered by a smaller particle for LMT when compared to GLMT.

## 3  Method

The following subsections describe the methodology by which the calibration data were obtained using both the static fibres and a conventional monodisperse aerosol – Sect. 3.1 and 3.2 respectively. While the instruments being calibrated were all UCASS units, there were two separate TIA gains, and therefore two separate size ranges which were approximately 0.4 μm to 18 μm for high gain (HG) and 3 μm to 40 μm for low gain (LG). As discussed further in Sect. 3.1, selection of fibres with low SCS values was non trivial, and three datapoints for each instrument were required. Therefore, the fibre calibrations for the HG UCASS units were conducted using two fibre SCS datapoints and one aerosol SCS datapoint.

### 3.1  Fibre Calibration

An OPS, be it naturally aspirated or aspirated via the means of a pump or fan, requires the movement of a transport fluid – usually air – through a sample area in a laser beam, in order to measure the optical properties of the particles contained within said transport fluid. Implicitly this means that any given particle trajectory will generate a temporal response, the amplitude of which is proportional to the SCS of the particle at any given point as it traverses the laser beam. While some OPSs record the



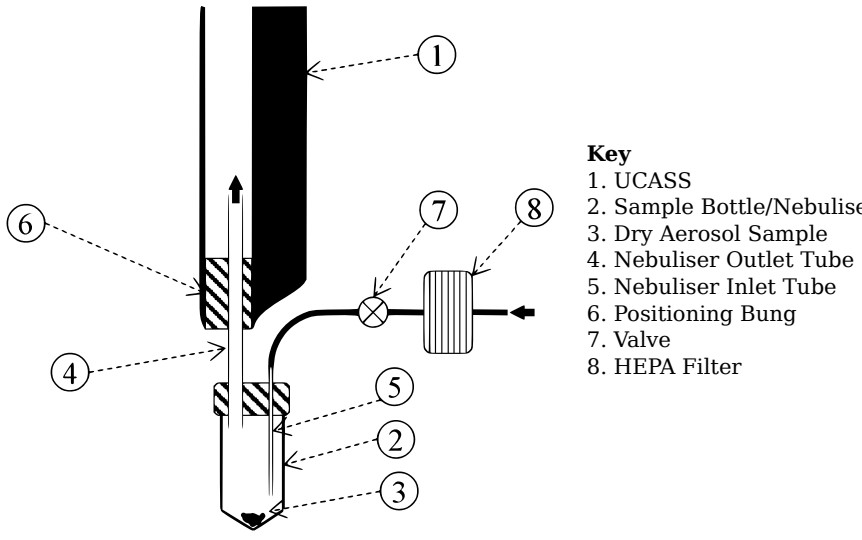

**Figure 5.** A diagram of the aerosol calibration apparatus used. This figure was taken from Girdwood (2023), and modified for readability and formatting.

whole temporal response of a particle for analysis and debugging purposes – and some will simply record the peak height of the particle, which is proportional to the SCS when it is in the centre of the beam – an OPS will always require this temporal response on the photodetector in order for the firmware to appropriately register a particle and record its optical parameters.

While considering a statically mounted fibre, there are a number of methods to potentially accomplish a temporal response similar to a real particle. The methods considered in this paper were: i) devising an apparatus to vibrate the fibre through the sample area of the beam; ii) directly measuring the output of the TIAs which are used to amplify the photocurrent from the detector into a voltage proportional to the SCS; and iii) devising an apparatus to flash the laser in a way which mimics the intensity profile of a particles temporal response. (i) is the most complex and would require significant development. A similar

technique was developed for use with optical array probes in Connolly et al. (2007); O'Shea et al. (2021), where ice crystal analogues were moved through the sample volume on a glass slide. This would likely not work for OPS type probes since the microscope slide would have a significant effect on the scattered light, and the van der Waals force would not be sufficient to properly attach the sampled medium to the slide for the accelerations required to reach the considered velocity range. In addition, this technique would be less suited for use in a field calibration tool, since it would likely require instruments to be

rigidly mounted to an optical bench.

(ii) is the simplest approach, since this would not require any additional apparatus aside from a multimeter to measure the TIA output. However, the method requires a minor modification to potentially expensive equipment – that is, flying leads attached to the TIA output pins – and a replication in the various computations conducted in both the instrument analogue



electronics and firmware in order to calculate a scaled instrument response from a voltage, which are often opaque. In addition,
the TIAs in some instruments may not be accessible.

(iii) can be accomplished via two means: using a rotating disc with a mask in the laser path to flash the laser at particular intervals, or electrically manipulating the laser driver circuit in order to turn the laser on and off at a specific frequency. The latter has the benefit of not involving the use of mechanical components beyond the introduction of the fibre to the laser beam, in addition to the potential to add a laser flashing mode to the instrument firmware for calibration purposes. Unlike (ii), this
method has the additional benefit of utilising the same computation and pulse filtering steps which are built into the instrument, meaning that if a calibration pulse is measured in the instrument software, it would be counted as a valid particle. In this instance, method (iii) via the means of electrical manipulation of the laser driver circuit was chosen, since any method of mechanically masking the laser would not fit in the space in the UCASS beam optics assembly. However, this would be an interesting method to test with different instruments.

In order to flash the laser with a Gaussian pulse, the laser driver had to be modified to minimise the input capacitance. This was because the maximum pulse width of the temporal response accepted by the UCASS is 50 µs, which was filtered out by the standard circuit. Hereafter the modified circuit will be referred to as the laser pulse driver (LPD). The major requirement for the LPD was to be able to flash the laser with a pulse width which would both be accepted by the instrument firmware, and have a peak pulse irradiance equal to the static irradiance of the continuous wave beam. In order to accomplish this, all
the filtering capacitors were removed from the LPD, and a low-side switch was added to the circuit via the addition of an enhancement mode N-channel MOSFET, the gate of which was driven, in this instance, by a signal generator which was set to produce a Gaussian pulse. A future development of this design could negate the use of an LPD entirely by utilising a version of the instrument firmware which could drive the laser with a pulse, in addition to a different choice of filtering capacitors in the laser driver circuit, which were originally oversized since the laser only needed to achieve continuous wave operation.

The differential SCS of the statically mounted fibre depends on both its radius and refractive index. For this prototype study, it was determined that three fibre SCS data-points would be sufficient, since the relationship between analogue to digital converter peak height and scattering cross section is linear, and can therefore be modelled with a two-coefficient polynomial. However, for future studies, more data-points surrounding the Mie oscillations in the particle diameter lookup table should be obtained, while surmounting practical limitations surrounding the manufacture of the fibres themselves. The three fibres were
chosen to have a similar SCS to that of their aerosol particle counterpoints, which will be discussed further in Sect. 3.2. One drawback to this calibration method was the difficulty of obtaining a fibre with a SCS small enough to obtain a calibration datapoint in the lower end of the OPS size range. This is because the fibre spanned the entire width of the laser beam minor axis, thus resulting in a larger exposed area when compared to a sphere of similar diameter. In order to surmount this issue, fibres constructed from materials with large imaginary refractive index components were chosen, these were: tungsten and
extruded carbon. A summary of the chosen fibres in ascending order of SCS magnitude is provided in Table 1, and microscope images of said fibres are presented in Fig. 4. Future expansions of this research would benefit from advanced ultrafine fibre manufacturing techniques.



**Table 1.** A summary of the fibres used as calibration data points. The given refractive index is calculated using the wavelength of the UCASS laser, which is 658 nm.

| Label | Material | Diameter (µm) | Refractive Index |
|-------|----------|---------------|------------------|
| f1 | Carbon | 6.0 | 2.514+0.02j |
| f2 | Tungsten | 13.0 | 3.652+2.80j |
| f3 | Borosilicate Glass | 5.5 | 1.515+0j |

**Table 2.** A summary of the aerosol particles used as calibration data points. The given refractive index is again calculated using the wavelength of the UCASS laser, which is 658 nm.

| Label | Material | Diameter (µm) | Refractive Index |
|-------|----------|---------------|------------------|
| fa0 | Silica | 2.00 | 1.440+0j |
| a1 | Silica | 5.50 | 1.440+0j |
| a2 | Borosilicate Glass | 11.58 | 1.515+0j |
| a3 | Borosilicate Glass | 25.58 | 1.515+0j |

In order to mount the fibres in the sample volume of the UCASS, a custom mount was designed – these are shown in Fig.
3. The mounts were constructed from 3D printed ABS, and had two components. Each end of the fibres was positioned in a groove in each of the components, which were fitted together in such a way as to be able to stretch the fibre tight once it was in position. A locking screw was added to secure the two parts of the fibre holder at a fixed distance apart. In order to position the fibres in the mounts, a microscope was used with a micromanipulator. Once the fibres were in the mounts, hot glue was used to secure both ends in place, then the fibre could be pulled tight in the holder to ensure it intersected the sample volume
without flexing. Hot glue was chosen as the adhesive, since the fibres were not under particularly strong force, and the adhesive could be easily removed without damaging the fibre holder should the fibre snap during its mounting process. As a short test of the validity of this method, the fibre was moved around in the UCASS sample area while the LPD was turned on. This was to ensure that the response of the instrument did not change much depending on the fibre position. If the response of the UCASS instruments varied too much, this was considered to be a flaw in the instrument optical alignment – which could cause an
anisotropic distribution of light on the photodetector – or a highly variable laser energy density distribution across the sample volume. This was not the case for any of the UCASS units which were used for the experiments presented in this paper.

### 3.2  Aerosol Calibration

The aerosol calibration method was similar to that described in Rosenberg et al. (2012); Smith et al. (2019). Small amounts
of dry monodisperse glass beads and silica microspheres were placed in a nebuliser, which was flushed with clean air from a compressor through the instrument. Since the UCASS is an open path instrument, the airflow velocity at the outlet of the





nebuliser needed to be measured in order to ensure that the aerosol particle beam traversal times are within the range of acceptance values defined in the instrument firmware. A summary of the calibration aerosol particles used is presented in Table 2, and microscope images of said aerosol particles are shown in Fig. 4.

A diagram of the apparatus can be found in Fig. 5. The air from the compressor, was filtered with a HEPA filter to minimise contamination from external sources. The air behind the system valve was pressurised to 2 atmospheres to ensure enough turbulence was generated in the bottle in order to break up the particles, and minimise the presence of doublets and triplets; residual doublets and triplets were present as obvious secondary and tertiary peaks, and hence filtered out in post-processing. The exact pressure needed to accomplish this was found via an iterative process, since identification of doublets and triplets in

the data was trivial. The nebuliser consisted of a sample bottle with an inlet for clean air, and an outlet for sample-laden air. Sample was inserted into the nebuliser by the means of a sterile scalpel blade, which was disposed of between calibrations. The calibration for one size of aerosol was conducted for all instruments, after which the apparatus was cleaned thoroughly. Clean air was then run through the apparatus with a high gain UCASS measuring the output, in order to ensure it was clean.

## 4   Results and Discussion

The per-unit results for the fibre and aerosol calibrations for both the high-gain and low-gain UCASS units are shown in Fig. 7 and 6 respectively. These plots show the SCSs of the calibration data – and corresponding approximate water droplet radius on the top axis – against the instrument response of each UCASS unit to said datum. The datums used are labeled with their corresponding table entry in either Table 2 or 1 at the top of each figure. In Fig. 7, the datum labeled as 'fa0' is the one which has been shared between the two calibrations. The serial number of each UCASS unit is annotated the corresponding plot panel.

The theoretical response – which is computed from idealised values for TIA current gain, alignment, and laser power – of each UCASS unit is denoted by the dotted line. This is mostly shown for reference, since the deviation of any one calibration from this line would be indicative of the performance of an individual unit, as opposed to being indicative of a calibration related issue.

    It is obvious at this stage that the low-gain UCASS calibration was far more successful than that of the high-gain UCASS.

Calibration datum 'f1' produces a consistently larger-than-expected instrument response for all of the high-gain units. This is likely not an issue with the computation of the 'f1' datum SCS, since the effect is not present for the low-gain units, where the same calculation parameters were used. Since the main difference between the two gain modes is the inclusion of the 'fa0' datum, which is at the lowest extremity of the size range, it is probable that the error results from a discrepancy in the measured offset – coefficient c0 – between the two calibrations. The calibration computations utilising the fibre data were consistently

larger than that which utilised the aerosol data; the correlation of fibre 'c0' to aerosol 'c0' resulted in a regression gradient of 2.7, with a correlation coefficient of 0.6, which is shown in Fig. 8a. Throughout Fig. 8, the 'x' markers show a value from a low-gain UCASS, and the 'o' markers show a value from a high-gain UCASS.

    Deviations in the offset coefficient result from constant artefacts which affect all particle sizes equally. Mostly, this is due to a non-zero static stray light resulting from ineffective termination of the laser in the beam dump, or diffracted light from the 2





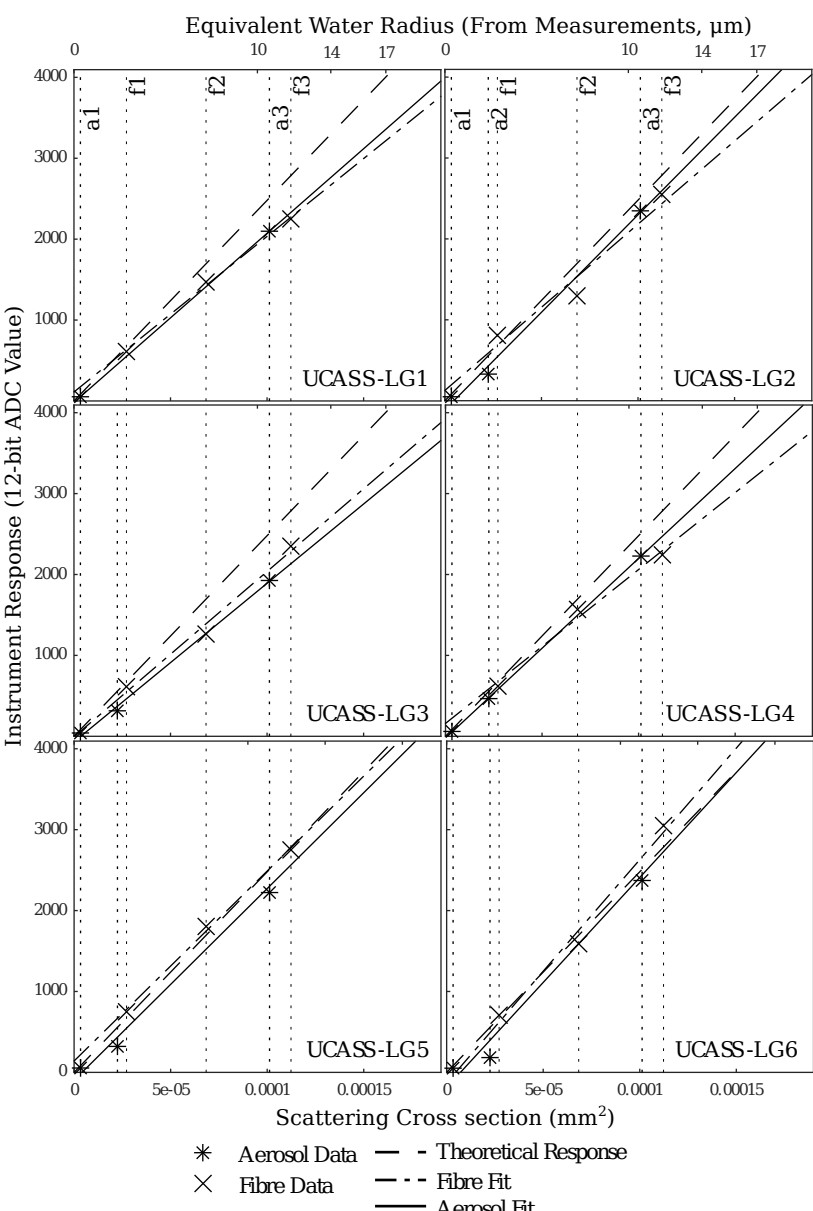

**Figure 6.** Overview of the calibrations for the low-gain UCASS units. The instrument response is on the y-axis and scattering cross section is on the bottom x-axis. The top x-axis on all the plots is a log axis and shows to the approximate radius of a water droplet with a scattering cross section corresponding to the lower x-axis. The vertical dotted lines highlight the calibration particles used, and the label corresponds to the table entry in Tables 1 and 2

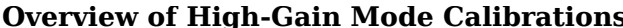

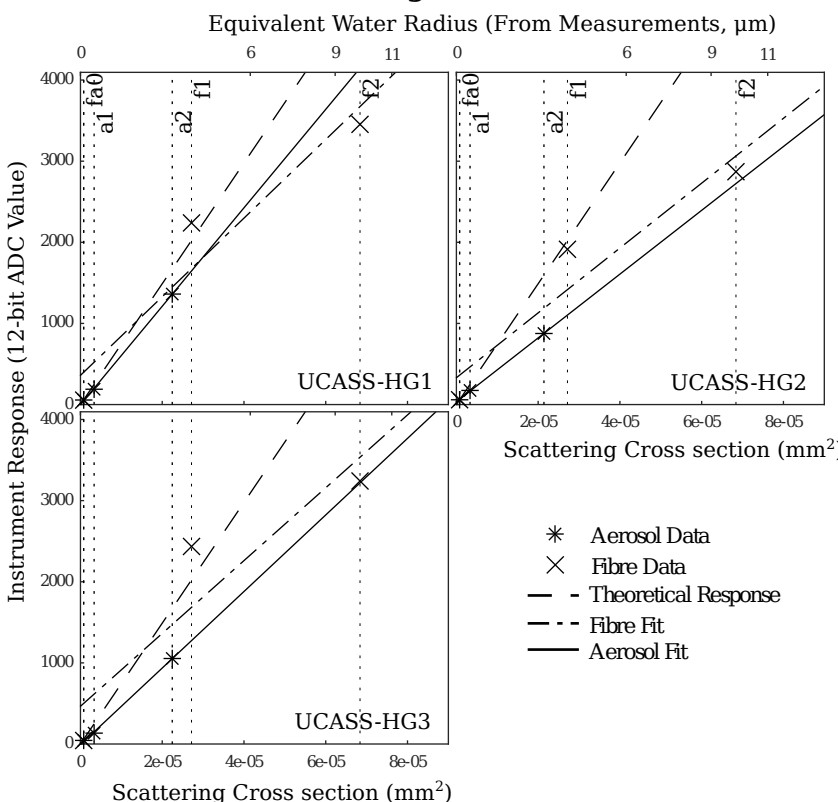

**Figure 7.** The same as Fig. 6, but for the high-gain UCASS units. The datum labeled 'fa0' is that which is shared between the fibre and aerosol calibration computations.

mm aperture in the beam forming optics. Differences in the offset coefficient can also result from the tolerance in TIA internal voltage offset. Since the estimation of the offset coefficient for the aerosol calibration is often negative, which is impossible, part of this issue is likely due to an error in the calculation of the SCS projected by the spherical particles. The stray light value could also have been higher for fibres due to the fibre mount reflecting stray laser light onto the collecting optics and the detector. Figure 8d shows the correlation of the instrument photocurrent gain – coefficient 'c1' – with the offset coefficient 'c0'. This plot should reveal how much of the offset coefficient resulted from stray light, since photocurrent gain will affect stray light and light scattered from a particle in the same way. The gradient of the regression line for the fibre calibration was around twice that for the aerosol calibration, thus indicating more stray light present in the fibre calibration. However, the correlation coefficient was small in both cases, which indicated another influencing factor on the stray light discrepancy.

It was considered that this additional factor resulted from the use of GLMT simulations, some errors in computation of the theoretical SCS of a calibration particle would influence the calibration curve. However, as was demonstrated in Fig. 2, the difference between GLMT and LMT was mostly only significant for particles with larger diameters when compared to the



**Correlation Plots**

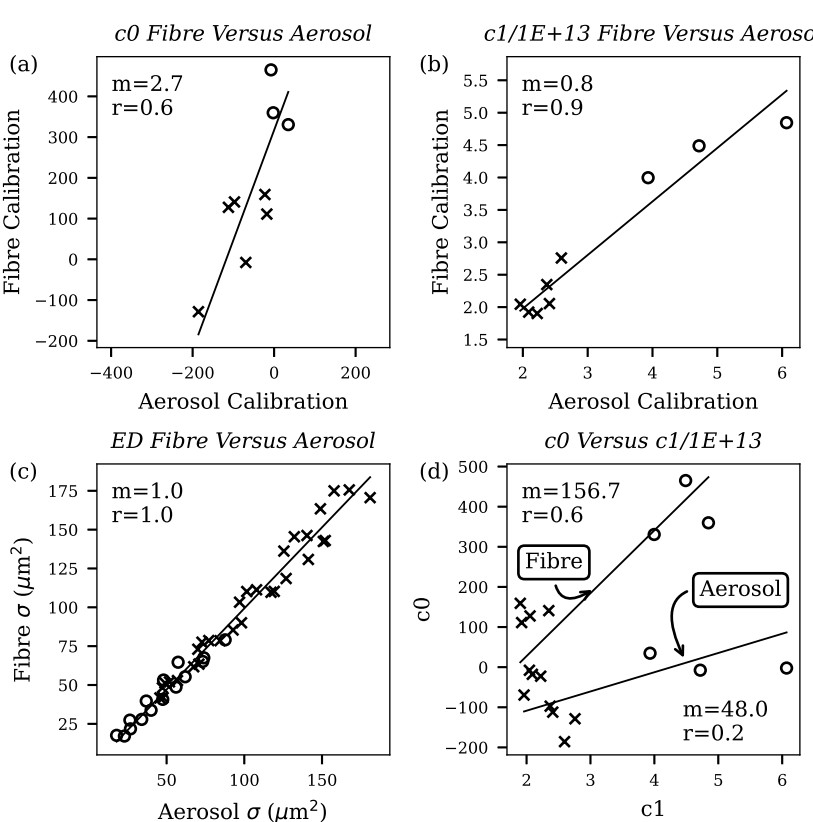

**Figure 8.** Correlation plots for all of the calibrations. (a) shows the $0^{th}$ coefficient for measured for the fibre and aerosol calibrations; (b) shows the same but with the $1^{st}$ coefficient; (c) shows the difference in effective diameter computed from a simulated Gaussian distribution with unity standard deviation, with varying initial effective diameter; and (d) shows the $0^{th}$ coefficient correlated with the $1^{st}$ coefficient for both calibrations.

beam width, and the beam width of the UCASS is 40 μm. Instead, the negative results for the offset coefficient likely resulted from a misrepresentation of either the modal size or refractive index of one of the mono-disperse samples used. Since Fig. 7 and 6 showed a systematic overestimation of the SCS of calibration particle 'a2', it is likely that this particle standard in particular 270 was the cause. This highlights one of the difficulties of calibrating with mono-disperse aerosol, since the calculations are highly sensitive to deviations in the modal diameter and refractive index of the particle standards used, and manufacturing techniques for said samples are subject to error, which results in a tolerance stack. Figure 4 shows the calibration standards which were used in these experiments. The deviation in size between different particles is apparent particularly for the 'a2' standard, which was unexpected in a monodisperse sample. It is difficult to determine the mode of this distribution from a microscope image,





however the broadness of the sizes which were revealed offer once possible explanation as to why this standard in particular
was overestimated in diameter.

Despite this, the correlation of the photocurrent gain coefficient – 'c1' – computed from the fibre and aerosol data was
reasonable, 0.9, and had a gradient of 0.8. This correlation is shown in Fig. 8b. This indicated that the aerosol data yielded
slightly higher photocurrent gain coefficients than the fibre calibration on average. This discrepancy could be due to an incorrect

measurement of mono-disperse particle modal diameter at the factory, or an incorrect measurement of the fibre diameter under
the microscope. The fact that the 'c1' correlation was reasonable but 'c0' was not indicated that the error in the offset was
systematic across all of the particle standards. This, again, did not indicate that the error was due to a discrepancy between
LMT and GLMT simulations, since larger sizes would be affected more strongly, causing a discrepancy in both coefficients.
The systematic error could be the result of manufacturing of the calibration standards, however more information would be

needed to validate this claim, and part of this offset can be explained by the afore mentioned stray light reflected off the fibre
holder. Since the agreement between the two calibrations depended on many factors which were unlikely to ever be perfect, it
is important to know the sensitivity of the final data to these errors.

In order to demonstrate the sensitivity of the predicted effective diameter of a distribution of particles to the two coefficients,
a number of normal distributions of particles were simulated, then the two calibrations were compared with one another. The

normal distributions had unity standard deviation; the initial mean was varied between 12-bit ADC values of 500 and 3500,
with 5 data-points in total, which were then binned into 16 bins (the number of bins which the UCASS is capable of reporting.
The aerosol and fibre calibration data were then used to compute the "measured" size from the simulated ADC histograms.
The effective diameter – that is, the ratio of the third to second statistical moments as defined in Korolev et al. (1999) – of the
size distributions was chosen as an appropriate dependant variable to compare, since this is a commonly used parameter. Since

diameter is not computed from the SCS in this instance, the parameter used is actually the effective SCS. The results from this
experiment are presented in Fig. 8d, where it can be seen that the gradient of the best fit line here is 1.0, and the correlation
coefficient is 1.0. This result demonstrated that the measured effective SCS was not particularly sensitive to discrepancies in
the two coefficients at this magnitude

## 5  Conclusions

A calibration methodology for OPSs was devised utilising statically mounted fibres as the scattering media. An equation for the
SCS of a statically mounted fibre was derived from Bohren and Huffman (1998), along with an instrument generic weighting
function – Equation 8. Calibration data were obtained from nine UCASS instruments (Smith et al., 2019), using both static
fibres, and a conventional mono-disperse aerosol technique. For the fibre calibration, the fibres were positioned on custom-
designed mounts, and the laser beam of the UCASS was pulsed so as to mimic the temporal response of a particle trajectory.

For the aerosol calibration, GLMT simulations were used in place of conventional LMT to account for the shape of the laser
beam minor axis at the sample volume. Since the relationship between instrument response and scattering cross section – that
is, the response of a TIA-detector circuit to a given light input – is linear, the calibration can be conducted in this domain, and





only two regression coefficients were needed. The calibration data using both methods were compared and analysed, and the principle artefacts which affected said data were discussed. One technique which was adopted to aid the inter-comparison was simulated data, where a particle size distribution was created in the ADC response domain, which was then converted to SCS using the two calibrations. The principle conclusions of this work can be summarised as follows:

1. The use of LMT caused an under-estimation in particle SCS when the particle had a large diameter when compared with the beam waist. GLMT simulations were proposed as a solution to rectify this, however more research needs to be done on how different OPSs are affected.

2. The effective SCS response of the two calibrations to simulated data were in notable agreement, with a linear regression gradient of 1.0 and a correlation coefficient of 1.0. This was sufficiently close to conclude that the small artefacts which were encountered had little influence on the final result.

3. Reflections of stray light off the fibre holder caused a slight increase in the $0^{th}$ calibration coefficient. This was found to be one of the most significant design factors relating to the fibre mounting system.

4. Discrepancies in the $0^{th}$ calibration coefficient may have also been caused by a misrepresentation of the size or of the aerosol particles themselves, as was revealed by microscopy.

5. The low-gain UCASS calibrations agreed far better than the high-gain UCASS units, owing to the inclusion of a shared datum between the two calculations.

6. Selection of the fibres was important, and it was non-trivial to find a fibre small enough to cover the smallest extreme of the SCS domain, a stark contrast to conventional calibrations.

Future development work will be conducted by the authors into utilising the fibre calibration technique as a tool for calibrating instruments in the field. This is of particular interest to researchers who work on uncrewed aerial vehicles (UAVs) and conventional aircraft, since knowledge of how the calibration of an OPS has drifted between flights can be used to apply corrections to data during analysis. For this application in particular, the laser flashing method is ideal since the apparatus would be simple. However, The addition of an LPD to an instrument is non-trivial, and may be undesirable for expensive instrumentation on conventional aircraft. For this reason, flashing the laser by means of periodically blocking it mechanically should also be investigated. In addition, this technique could be applied to mass-produced, low-cost OPSs, since it would be more simple to deploy this device in a factory environment than a conventional aerosol technique. In tandem to this, future work will be conducted into manufacturing techniques for small fibres, since the fibre selection and characterisation is important. In particular, glass fibre stretching, and tungsten fibre etching are promising processes for the manufacture of fibre standards.

*Code and data availability.* For the purposes of the preprint manuscript, the data and code assets can be accessed from "https://github.com/wolkchen-cirrus/FibCalRepo".



*Author contributions.* Software was developed by JG and HB, writing was conducted by JG, resources and supervision were provided by CS and EH, conceptualisation was performed by JG, investigation was performed by JG and RL, and mothodology was developed by JG and RL.

*Competing interests.* The authors declare that they have no conflict of interest.



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
