# Peer review of "Calibration of Optical Particle Spectrometers Using Mounted Fibres"

_Atmospheric Measurement Techniques, 2024_

## Author Response (AR1)

**Author Response to RC1**

Jessica Girdwood -- jessica.girdwood@ncas.ac.uk

We thank the reviewer for their comments. Our response is as follows:

- In order to reach a more broad audience, we have added a paragraph in the introduction highlighting where optical scattering measurements fit into general aerosol measurement.
- While we agree that the OPC-N3 would be a good device to apply this calibration technique to, we were unable to get data with this instrument since the lab where I performed the original experiments was shut down while moving to a new building, and a new fibre mount would have had to have been developed. In addition, I am no longer at this institution, which adds another layer of complexity to this. However, the OPC-N3 uses exactly the same optical layout as the UCASS, since they were both branches off the same device (Kaye and Hirst 2015). This means that the calibration technique would implicitly be applicable to the OPC-N3, and I have stated this in the Conclusions section.
- The dash-dot line in figure 2 was changed to blue.
- The typographical error was corrected.

**Author Response to RC2**

Jessica Girdwood - jessica.girdwood@ncas.ac.uk

We thank the reviewer for their comments, and respond as follows:

- Line 205 -- abbreviation of ABS clarified as acrylonitrile butadiene styrene.
- Abbreviations in the abstract are now defined.
- Line 100 -- the stated assumptions describe an optical system where the proportion of light from a single diffraction fringe measured is independent of change in y-direction, which is true for the UCASS as determined experimentally. I have re-stated the assumptions to describe this directly.
- Line 114 -- the definition of sufficiently large might vary depending on the beam width of the instrument, and the definition of this is effectively the topic of this section. The manuscript was amended to state this.
- Line 134 -- This sentence makes more sense if the parenthetical information is within commas. Thank you for pointing this out.
- Line 135 -- "which" was removed.
- Line 180-189 -- The LPD was used in this case because we possess the knowledge to modify the chosen instruments, we would actually recommend to instrument manufacturers that they build this capability into their instruments if they wish to calibrate them this way. For a field calibration, a mechanism by which the laser is flashed using mechanical means would be far more attainable. A sentence at the end of this paragraph was inserted to clarify this.
- Swapped figures 3 and 4 around.
- Line 207 -- the fixed distance refers to a distance between the two parts of the fibre holder which was sufficient to cause no interference to neither the direct nor scattered light within the OPS. The manuscript has been amended to reflect this.
- Line 225 -- comma deleted.
- Figs 6 and 7 swapped.
- Since figure 6 is now 7 and vice versa, I will refer to these as the LG and HG figures respectively (after reorder). Both HG and LG fits were done with 3 data-points, thus mimicking a realistic factory calibration for a low-cost instrument. For the HG fits, an aerosol datum was incorporated into the fibre data, because a fibre with a small enough diameter could not be sourced. This is the datum which is labelled fa0. This was explained at the beginning of the method section (3), but I have re-stated this in the discussion section to avoid further confusion.
- Line 339 -- corrected to "methodology"